# Overview and Evaluation of Existing Guidelines for Rational Antimicrobial Use in Small-Animal Veterinary Practice in Europe

**DOI:** 10.3390/antibiotics10040409

**Published:** 2021-04-09

**Authors:** Fergus Allerton, Cameron Prior, Arzu Funda Bagcigil, Els Broens, Bénédicte Callens, Peter Damborg, Jeroen Dewulf, Maria-Eleni Filippitzi, Luís Pedro Carmo, Jonathan Gómez-Raja, Erez Harpaz, Ana Mateus, Mirja Nolff, Clare J. Phythian, Dorina Timofte, Flavia Zendri, Lisbeth Rem Jessen

**Affiliations:** 1Willows Veterinary Centre and Referral Service, Highlands Road, Shirley, Solihull B90 4NH, UK; 2Veterinary Specialists Scotland, 1 Deer Park Road Livingston, Scotland EH54 8AG, UK; Cameron.Prior@vetscotland.co.uk; 3Department of Microbiology, Faculty of Veterinary Medicine, Istanbul University-Cerrahpaşa, 34320 Avcılar, Istanbul, Turkey; fucigil@iuc.edu.tr; 4Faculty of Veterinary Medicine, Department of Biomolecular Health Sciences, Utrecht University, Yalelaan 1, 3584 CL Utrecht, The Netherlands; E.M.Broens@uu.nl; 5Centre of Knowledge on Antimicrobial Use and Resistance, Galileelaan 5/02, 1210 Brussels, Belgium; benedicte.callens@amcra.be; 6Department of Veterinary and Animal Sciences, University of Copenhagen, Stigbøjlen 4, 1870 Frederiksberg, Denmark; pedam@sund.ku.dk; 7Veterinary Epidemiology Unit, Faculty of Veterinary Medicine, Ghent University, Salisburylaan 133, 9820 Merelbeke, Belgium; jeroen.dewulf@ugent.be; 8Veterinary Epidemiology Unit, Sciensano, 1050 Brussels, Belgium; Maria-Eleni.Filippitzi@sciensano.be; 9Veterinary Public Health Institute, Vetsuisse Faculty, University of Bern, Schwarzenburgstrasse 161, 3097 Liebefeld, Bern, Switzerland; luis.gomesdocarmo@vetsuisse.unibe.ch; 10FundeSalud, Government of Extremadura, Pio Baroja 10, 06800 Mérida, Spain; Jonathan.gomez@fundesalud.es; 11Faculty of Veterinary Medicine, Institute for Production Animal Clinical Science, Norwegian University of Life Sciences, Small Ruminant Research and Herd Health, Høyland, 4325 Sandnes, Norway; erez.harpaz@nmbu.no (E.H.); clare.phythian@nmbu.no (C.J.P.); 12Hawkshead Campus, Royal Veterinary College, University of London, Hawkshead Lane, Hatfield AL9 7TA, Hertfordshire, UK; amateus@rvc.ac.uk; 13Clinic for Small Animal Surgery Tierspital Zürich, Vetsuisse Faculty, University of Zürich, Winterthurerstrasse 260, 8057 Zürich, Switzerland; mirjachristine.nolff@uzh.ch; 14Department of Veterinary Anatomy, Leahurst Campus, Institute of Infection, Veterinary and Ecological Sciences, Physiology and Pathology, University of Liverpool, Neston CH64 7TE, UK; D.Timofte@liverpool.ac.uk (D.T.); Flavia.Zendri@liverpool.ac.uk (F.Z.); 15Department of Veterinary Clinical Sciences, Faculty of Health Sciences, University of Copenhagen, Dyrlægevej 16, 1870 Frederiksberg C, Denmark; lrmj@sund.ku.dk

**Keywords:** antimicrobial stewardship, antimicrobial resistance, guidelines, AGREE II, canine, feline

## Abstract

Antimicrobial stewardship guidelines (ASGs) represent an important tool to help veterinarians optimize their antimicrobial use with the objective of decreasing antimicrobial resistance. The aim of this study was to map and qualitatively assess the ASGs for antimicrobial use in cats and dogs in Europe. Country representatives of the European Network for Optimization of Veterinary Antimicrobial Treatment (ENOVAT) were asked to identify ASGs published in their countries. All collated ASGs updated since January 2010 containing recommendations on antimicrobial therapy for at least three conditions affecting different organ systems in cats and dogs underwent detailed review including AGREE II analysis. Out of forty countries investigated, fifteen ASGs from eleven countries met the inclusion criteria. Several critical principles of antimicrobial use were identified, providing a framework that should assist development of stewardship guidance. The AGREE II analysis highlighted several methodological limitations of the currently available ASGs. This study sheds light on the lack of national ASGs for dogs and cats in multiple European countries and should encourage national bodies to prioritize guideline development in small animals. A greater awareness of the need to use a structured approach to guideline development could improve the quality of ASGs in the future.

## 1. Introduction

In response to the rising threat to both human and animal health from multidrug-resistant (MDR) infections [1,2], there is an urgent need to adopt measures to preserve the efficacy of available antimicrobials. Since antimicrobial use (AMU) is recognized as a key driver of antimicrobial resistance (AMR) [3], any steps that can reduce unnecessary or inappropriate AMU should diminish the selection pressure for resistant bacterial strains. The vast majority of AMU in animals occurs on farms promoting development of AMR and creating an important potential reservoir of resistance genes [4]. A 35% decrease in aggregated overall sales (in mg/population correction unit) was documented in Europe from 2011 to 2018, reflecting stewardship efforts in agriculture [5]. In companion animals, the overall AMU is substantially lower compared to production animals and the sales of antimicrobial tablets (a surrogate measure of AMU in companion animals) accounted for 1.1% of the total sales in tons in 2018 although the proportion was higher (5.2–10.3%) in some countries [5].

However, the use of critically important antimicrobials belonging to the European Medicines Agency (EMA) AVOID USE and/or RESTRICT USE categories may be more common in companion animals compared to production animals [6]. For example, in a food-producing country such as Denmark, companion animals account for the majority of veterinary fluoroquinolone consumption despite representing only 1% of the total veterinary AMU. Ignoring companion animals in the One Health equation on AMR may prove to be unwise as multidrug-resistant bacteria including methicillin-resistant *Staphylococcus aureus* (MRSA) [7,8], vancomycin-resistant enterococci (VRE) [9,10], extended-spectrum beta-lactamase (ESBL), AmpC and carbapenemase-producing Gram-negative bacteria [9,11,12,13] have been isolated in dogs and cats. This represents an important potential reservoir of resistance genes; transmission to susceptible bacteria could pose a serious risk to human health.

Encouraging rational AMU is a fundamental principle of antimicrobial stewardship programs (ASPs). In veterinary practice, a range of approaches have been adopted across Europe including the imposition of regulations restricting the use of specific antimicrobials and the promotion of voluntary antimicrobial stewardship guidelines (ASGs) at the national and clinical levels [14,15,16,17,18,19,20,21]. Restrictive policies limiting the use of particular antimicrobials in food-producing animals are already in place in several European countries and will come into force across the European Union in January 2022 [22]. In contrast, antimicrobial stewardship in companion animals in Europe relies more heavily on AMU guidelines, with few legislative measures which are mainly limited to Northern European countries.

To the authors’ knowledge, the only previous review of the main veterinary prudent use guidelines was performed in 2012 and included only those published in English [23]. The availability and quality of ASGs for dogs and cats in Europe have not been recently investigated.

The European Network for Optimization of Veterinary Antimicrobial Treatment (ENOVAT) is a European Cooperation in Science and Technology (COST) Action—a network dedicated to scientific collaboration with special emphasis on the development of ASGs and refinement of microbiological diagnostic procedures. A primary objective of the ENOVAT is to map and compare the availability, structure, and evidence base of veterinary ASG in Europe. This study focuses on national guidelines in Europe pertaining to cats and dogs only. By characterizing common features of published antimicrobial guidelines, it may be possible to build a framework of key recommendations that can inform other national organizations that are in the process of developing their own strategies. The Appraisal of Guidelines for Research and Evaluation Instrument (AGREE II) [24] was developed to address the issue of variability in the quality of best practice guidelines and has previously been applied to surgical prophylaxis guidelines in veterinary medicine [25]. The AGREE II instrument was incorporated into this study to offer an objective assessment of the identified guidelines, and to highlight aspects of the guideline preparation process that may be of relevance to future developers.

## 2. Materials and Methods

### 2.1. Identification of Potential Guideline Documents

Representatives of every country participating in the ENOVAT COST Action were contacted by email to inform them of this study and to seek assistance in the identification of ASGs offering recommendations pertaining to cats and dogs. Country representatives were asked to identify ASGs for cats and dogs published in their countries. In addition, they were invited to perform an internet search to find published ASGs, with searches to be performed in English and all official languages of the respective country. Search terms were ‘antimicrobial’ or ‘antibiotic’ or ‘antibacterial’; and ‘stewardship’ or ‘guidelines’ and ‘cat’ or ‘dog’ or ‘companion animal’ or ‘veterinary’. A further search verification step involved country representatives contacting their respective, national veterinary organizations (particularly those engaged in companion animal work) to assist in providing this information. Details of all identified ASGs were collated; the absence of appropriate guidelines after completion of the searches was also recorded. A centralized database of ASGs (the global repository of available guidelines for responsible use of antimicrobials in animal health) hosted by the World Veterinary Association [26] was also checked.

All ASGs, published or updated since January 2010, in any of the countries represented in the ENOVAT were eligible. If more than one version of any guideline was available in this timeframe, only the latest update was evaluated. To be included in the descriptive analysis, the guideline had to provide recommendations on the empirical use of antimicrobial therapy for at least three conditions affecting different organ systems in cats and dogs. A recommendation, per the World Health Organization Handbook for guideline development, implies a choice between different interventions (e.g., the decision to use antimicrobials) that have an impact on patient health and wider implications for the use of resources [27].

### 2.2. Evaluation of Identified Antimicrobial Stewardship Guidelines

Eligible guidelines were reviewed, either in their original language or after translation into English via an online translation tool (Google Translate) by two of the authors and responses were compared. All discordant results were verified and discussed to ensure a consensus was reached. Qualitative descriptors of each guideline document were recorded including the format of the document (book, poster, digital application/tool or document); the type of group responsible for the production of the guidelines (government body, national organization, society, university or an interested group); accessibility online; paid or free access to guidelines; what other species were included in the guidelines; and the languages in which the documents were available.

Specific features of each guideline document were reviewed including the number of organ systems for which recommendations were available; whether general principles of responsible AMU were included; whether common bacterial pathogens were listed for each infection; whether the use of appropriate diagnostic techniques for aseptic sample collection and for the identification of putative organisms (e.g., bacterial culture or cytology) were promoted and whether a mechanism for providing feedback to the guideline authors was incorporated.

The guidelines were also evaluated to see whether they contained advice not to administer antimicrobials in particular conditions including acute diarrhea, feline lower urinary tract disease (FLUTD), subclinical bacteriuria or peri/post-operatively for certain classifications of surgical procedure (clean +/− clean contaminated surgical wound class). The guidelines were assessed to see whether they included recommendations to preferentially select narrow rather than broad-spectrum antimicrobials; suggested doses and durations of antimicrobial therapy; contained warnings about potential adverse effects associated with certain antimicrobials; and incorporated recommendations to monitor AMU and local resistance patterns. The specification of particular antimicrobials that should not be used in animals and of antimicrobials classified as highest priority critically important antibiotics (HPCIAs), the inclusion of advice on management of multidrug-resistant (MDR) bacterial infections and consideration of zoonotic risk from particular pathogens were also evaluated.

The AGREE II instrument [24] was used to critically evaluate six domains, covering the scope and purpose of the guidelines, stakeholder involvement, rigor of development, clarity of presentation, applicability and editorial independence, and an overall guideline assessment. Each element was rated on a 7-point Likert-like scale from strongly disagree (1) to strongly agree (7). Individual domain scores were calculated by summing individual item scores in each domain and scaling the total as a percentage of the maximum possible score for that domain. The scaled domain score was calculated as Equation (1):(1)$$Domain\;score=100\times \frac{Obtained\;score-Minimum\;possible\;score}{Maximum\;possible\;score-Minimum\;possible\;score}$$

Self-directed training material available on the AGREE Trust website (https://www.agreetrust.org/resource-centre/agree-ii/agree-ii-training-tools/, accessed on 1 March 2021) was completed by each appraiser. Each guideline was appraised using the AGREE II instrument by four veterinarians from a panel of nine vets with an interest in antimicrobial stewardship and proficiency in English and at least one of the other languages of the selected guidelines. Appraisal data were collated and median domain scores calculated for each guideline. Domain results were evaluated separately and no pre-defined quality thresholds were set. Interrater reliability was assessed by calculation of intraclass correlation coefficient (ICC) using a one-way random-effects model and average measures. SPSS Statistics Software version 26.0 was used to calculate ICC per item and 95% confidence intervals. An ICC from 0.40 to 0.59 was considered fair correlation, 0.60 to 0.74 was considered good correlation and 0.75 to 1.0 was considered excellent correlation [28].

## 3. Results

Representatives of 40 countries were contacted to provide information about antimicrobial guidelines developed or used in their respective countries (Figure 1). Replies were obtained relating to 38/40 countries, identifying 23 potential guidelines from 17 different countries, a further pan-European document produced by the Federation of European Companion Animal Veterinary Associations (FECAVA) and the Guidance for the rational use of antimicrobials (GRAM) book. Ten guidelines did not meet the study inclusion criteria either because the documents were produced for owners rather than vets (n = 1), addressed dogs only (n = 1), presented lists of conditions for which antimicrobials could be used rather than proposing antimicrobials for specific conditions (n = 3) or provided general stewardship advice only without including specific antimicrobial recommendations for clinical conditions (n = 5). Fifteen ASGs were identified for detailed review (Figure 1 and Table 1).

The eligible ASGs were available in different formats—six were online documents (41–221 pages in length), five were books (ranging from 56 to 560 pages in length), two were posters and two were online tools. The two resources developed in Switzerland may be considered as complementary tools and were produced by the same collaboration. They have been evaluated separately here as, individually, they met the inclusion criteria for this study. A range of different groups have been involved in development of ASGs including representatives from Universities, governing bodies, veterinary societies, a veterinary pharmaceutical firm and independent organizations. Direct involvement of a government agency was not recorded for any of the 15 ASGs. Eight of the 15 ASGs provide recommendations for cats and dogs only. Recommendations pertaining to a broad range of other species, including ruminants (cattle, sheep or goats), pigs, horses, fish, poultry, bees, rabbits, rats, guinea pigs, hamsters, tortoises, lizards, crocodiles, snakes, parrots and fur animals were also included in the other seven resources.

The primary language of the ASGs reflected the official language(s) of the relevant countries in which they were developed. Six of the fifteen ASGs were available in English (Table 1). The Danish guidelines have also been translated into Chinese, Polish and Slovene; the UK guidelines into Spanish, the GRAM book into French and Spanish and the FECAVA guidelines are available in Croatian, English, French, German, Lithuanian, Polish, Portuguese and Slovene. Fourteen of the 15 ASGs could be accessed online (one of which required payment to view and download, one required registration and creation of a user account) and 8/15 can also be requested as printed/hard copies (one printed book only available on payment).

A mechanism to provide feedback, in the form of a contact e-mail address, was included in 9/15 ASGs. Ten of the 15 guidelines reported that feedback on previous editions had been incorporated in the latest version. Since 2017, 11 of 15 ASGs have been updated, and new versions of three were in preparation. An accessible reference list is provided alongside 12/15 of the ASGs.

The ASG provided recommendations for antimicrobial selection for a varied number of conditions. These were separated into a median of 9.5 different organ systems (range 4–13). All of the ASGs included advice to restrict the prescription of antimicrobial medication where either non-bacterial disease or bacterial infections that may self-resolve without antimicrobial therapy is suspected. Recommendations not to prescribe antimicrobials for at least one specific indication were found in all of the ASGs (Table 2).

Specificities relating to AMU such as suggested doses, treatment durations and potential adverse effects were included in 11, 14 and 11 ASGs, respectively. A recommended treatment duration for uncomplicated urinary tract infections (UTI) was described in 13/15 ASGs including 3–5 days (n = 4), 7 days (n = 5) and 7–14 days (n = 4). The four ASG recommending treatment durations of 3–5 days had been updated in 2018 or later. Considerations for the management of MDR bacteria (e.g., MRSA and methicillin-resistant *Staphylococcus pseudintermedius* (MRSP), ESBL-producing Enterobacterales) were apparent in 10 of 15 ASGs. Nine of 15 ASGs contained recommendations to observe special precautions when dealing with potential zoonotic pathogenic bacteria.

### AGREE II Analysis

Median scores for the four appraisers for each AGREE II item and all 15 ASGs are shown in Figure 2.

Median domain scores are shown in Figure 3. Overall intraclass correlation coefficient amongst reviewers in this study was excellent (ICC = 0.80). Interrater reliability was individually calculated for each item rating and shown in Table 3 with scores ranging from poor correlation (−0.25 and 0.02) for items 8 and 20 to good correlation (0.71 and 0.70) for items 4 and 13, respectively.

## 4. Discussion

This study identified 15 antimicrobial stewardship guidelines on rational antimicrobial use for cats and dogs from 11 of the 40 countries investigated, highlighting a substantial gap in national recommendations for small-animal antimicrobial stewardship across Europe. None of the included guidelines were initiated by governmental initiatives, further emphasizing that small animals constitute a blind spot in the national AMU political agenda.

Antimicrobial use guidance produced in six other countries did not meet the inclusion criteria of this study (including one document produced for dogs only), several guidelines have been translated into multiple languages and 13/15 of the appraised ASGs are freely available online transcending national borders. ASGs from outside Europe have not been considered in this study; absent specific-country representatives, it was not possible to assuredly locate all ASGs in these territories. Topic-specific guidance has been disseminated by the International Society for Companion Animal Infectious Diseases (ISCAID) in the form of consensus statements [44,45,46,47], affording broader accessibility to stewardship advice. Guidelines produced by the FECAVA and the GRAM book were included in this analysis given their availability across Europe. However, national guidelines confer advantages as they can take into account local disease prevalence, AMR occurrence, AMU trends, any national regulations relating to antimicrobial prescription and the availability of antimicrobial formulations and diagnostics.

Key recommendations featured consistently across the majority of appraised ASGs, indicating adherence to a responsible AMU agenda within Europe, despite national differences. Eliminating inappropriate AMU for common, and typically benign and self-limiting, indications seems a logical and achievable target; the appearance of variants of the ‘do not use’ recommendations in all of the ASGs emphasizes the importance of this approach. The majority (14/15) of ASGs also outlined appropriate non-antimicrobial therapies—a means to avoid veterinarian frustration and to provide suitable alternative therapies that may address prescription pressure from the owner [48,49,50].

All of the ASGs encourage pathogen identification via cytology and/or bacterial culture to guide treatment decisions. Although bacterial culture with antimicrobial susceptibility testing (AST) is considered an essential stewardship tool [51,52], studies suggest infrequent application in small-animal practice in Europe often due to cost constraints [53,54,55]. This disappointing dichotomy amply demonstrates the difficulty of converting a key message into implemented actions. AST facilitates optimal drug selection and de-escalation of unnecessarily broad-spectrum empiric antimicrobial therapy. The growing availability of molecular testing modalities (e.g., PCR) and matrix-assisted laser desorption/ionization-time of flight mass spectrometry (MALDI-TOF MS) will provide veterinarians with greater opportunity to rapidly identify putative pathogens [56] and may have the potential to change diagnostic patterns.

The preferential use of narrow-spectrum antimicrobials was another commonly (13/15) included recommendation among the ASGs. Despite recognition of the greater potential to contribute to AMR, the use of broad-spectrum formulations, such as amoxicillin-clavulanate, far outweighs narrow-spectrum alternatives in Europe [57,58]. Here, external factors (e.g., drug familiarity, practice purchasing policy) are likely stronger influences in drug selection than guideline recommendations.

In a similar vein, all ASGs promoted topical rather than systemic AMU where appropriate. Given that skin conditions are among the most common motives for consultation in small-animal practice [59,60], a transition to topical use could significantly decrease total AMU limiting exposure of the gut microbiota to antimicrobials and reducing AMR. Interestingly, sequential studies in the UK (performed by different animal health surveillance networks) documented a drop in systemic antimicrobial therapy from 92% to 25% in dogs with skin disease [59,61]. The four-year gap between the data collection for these studies coincides with the launch of the first edition of national stewardship guidelines in the UK. Greater practitioner awareness of AMR may have contributed to these improved figures.

The inclusion of an explicit statement to reserve certain antimicrobials (carbapenems, linezolid and vancomycin) exclusively for human use in 12/15 of the ASGs serves as a vital reminder to practitioners who may not appreciate the public health importance of these agents. In many countries veterinary prescribers retain the autonomy to use these antimicrobials even though this raises serious ethical questions [62,63]. In Sweden, laws prohibit all veterinary use of these antimicrobials; similar restrictions may be introduced more widely. The World Health Organization (WHO) [64] and the European Medicines Agency [65] have independently categorized some antimicrobials as HPCIAs including fluoroquinolones, macrolides and third- and above generation cephalosporins. Imminent European legislation will recommend conditioning their prescription based on AST results [22]. To encourage vets to position HPCIAs appropriately in their treatment plan, many ASGs list them as highest-tier (last resort) options, stress the requirement for AST or use different color schemes (e.g., traffic lights) to underscore their restricted application.

Antimicrobial prescription (in terms of dose and duration of therapy) should adhere to the datasheet or Summary of Product Characteristics. There are instances when deviation from standard dosing is warranted based on patient factors or pharmacokinetic considerations (e.g., target organ penetration); highlighting these situations can ensure the efficacy of antimicrobial treatment. Shorter treatment courses decrease AMU and the risk of propagating AMR; regarding antimicrobial therapy duration, a shorter is better mantra has been endorsed in human medicine [66,67]. This approach has spilled over into veterinary medicine; the updated 2019 ISCAID guidelines on urinary tract disease [47] favor a shorter 3–5 day course for uncomplicated UTI compared to 7–14 days in the 2011 version [68]. The 3–5 day treatment recommendation has already been adopted by four of the most recently updated ASG reviewed in this study illustrating the value of regular revision of ASGs. There is a need for clinical trials to support this approach and move forward agendas which should feed into future guideline recommendations. Ominously, practitioners in human medicine often prescribe longer courses than the duration recommended in national guidelines [69]; additional measures may be required to support vets to adopt shorter antimicrobial courses, potentially including individualized treatment lengths linked to clinical resolution or surrogate markers such as C-reactive protein [70].

A lack of supportive clinical data represents an important limitation inherent to most recommendations included in small-animal ASGs. Recommendations are oft derived from expert opinion and extrapolation from human medicine principles. A similar situation was recognized in the early part of this century by WHO [71] prompting the adoption of rigorous and transparent processes that utilize the best available evidence [27,72]. Tools to assist guideline development have been incorporated into the process including the Grading of Recommendations Assessment, Development and Evaluation (GRADE) approach [73,74] and the AGREE Collaboration [75]. The AGREE II instrument [24] can also be used to critically appraise existing clinical practice guidelines and was employed in this study to review the identified guidelines.

Guideline evaluation was performed by appraisers with little prior experience of AGREE II. Raters completed the training modules provided on the AGREE Enterprise website but unfamiliarity with this tool is a limitation of this study. It is recommended that at least two, and preferably four, appraisers review each clinical guideline to increase reliability. However, nine ASGs were available in six different non-English languages. A wider pool of appraisers including at least one native speaker per ASG was sought to limit miscoding. Overall inter-rater reliability (IRR) was excellent, suggesting that less than 20% of the variability in the item scores was due to random variation. IRR for individual items was lower reflecting the smaller sample sizes but 15/23 items were classified as fair or good correlation. Poor agreement was found for rating items whose assessment required more subjective interpretation.

AGREE II Domain scores can be used to identify strengths and limitations of guidelines and provide an indication of their methodological quality. Median domain scores were highest for domain I, concerned with the overall aim of the guideline, the specific health questions tackled, and the target population, and domain IV, relating to the language, structure, and format of the guideline. These features are expected to score highly given the scope of the ASGs selected and the pre-determined inclusion criteria. The low median score for developmental rigor reflects a failure to detail methodological steps undertaken to evaluate the available evidence and to formulate recommendations. This information was frequently omitted from the ASGs, prohibiting consideration and prompting low scores. The retrospective application of the AGREE II instrument in this study imposes limitations as the ASGs may not have been designed with the AGREE criteria in mind, and it is possible that the true level of engagement with systematic processes is underestimated. Additionally the absence of adequate resources to exercise a systematic approach and the poor evidence base concerning AMU in small animals may favor a consensus-led approach to formulating recommendations.

Domain V covers aspects pertaining to guideline implementation and uptake that would routinely be included as Appendix A. Item 20, relating to resource implications, is not applicable; antimicrobial cost information will depend on multiple other factors outside the remit of ASGs. Within the AGREE II analysis, Appendix A can be consulted but was not routinely available to the appraisers in this study. The lowest median domain score related to editorial independence reflecting a failure to declare conflicts of interest or the role of any funding bodies. Nonetheless, these ASG have been developed by veterinarians, often contributing their time and expertise voluntarily. The low score recorded here is a reminder of the value of transparency; future guideline developers should include appropriate declarations.

Heterogeneity (exemplified by the wide range for some domain scores) was particularly high for domain II. In human medicine, stakeholder, especially patient, engagement is considered a necessary component of guideline development and implementation [76]. By making recommendations and setting outcomes that are relevant and important to key stakeholders, rapid meaningful change in antimicrobial prescribing behavior can be achieved [77]. In the context of these ASGs, stakeholders include the pet owner and the primary care veterinarians. Ensuring active involvement of different stakeholder groups has underpinned the successful Prescribing Champion Programme for Welsh veterinary farm practices [78]. Straightforward and cost-effective means to survey the views of stakeholder groups for veterinary guidelines will afford them a valuable influence. However, escalating methodological standards must be balanced to expected incremental benefits and should not present obstacles to guideline production [79].

At present, there are no empirical data to link AGREE II domain scores with guideline implementation outcomes. The ultimate success of any ASG is judged on its impact on total and specific antimicrobial usage and the degree of prescriber adherence. Frustratingly studies have shown a high frequency of antimicrobial prescription for conditions where AMU is routinely discouraged (e.g., acute diarrhea, acute vomiting, and acute upper respiratory tract disease) demonstrating that inappropriate antimicrobial prescription remains a significant issue [59]. ASG form part of an enablement approach to stewardship [14]. It is hoped that through sustained prescriber engagement, AMU can be optimized (e.g., shifting away from using critically important antimicrobials) and, ultimately reduced. However, the transfer of knowledge alone may be insufficient to influence prescriber behavior [80] and additional measures that address emotional, cognitive and interpersonal factors are warranted to effectuate meaningful change [81,82].

In people, the ability of ASGs to cause a decrease in AMU without adversely affecting patient outcome has been established in low- and middle-income countries as well as in hospital inpatient and outpatient settings in the USA [83,84,85]. Few studies have described the adherence to guidelines and/or the effect of guidelines on antimicrobial prescription habits among small-animal veterinarians. Awareness of small-animal guidance among prescribers in the UK has been demonstrated [86]. This may not automatically mean that the guidance is followed as adherence to recommendations was poor in both university hospitals and private veterinary practices in Switzerland [87]. More encouragingly, efficacy of hospital policy and national ASGs to positively influence prescription habits of small-animal practitioners has been shown in Canada and Denmark [55,88,89,90,91].

Improved access to AMU data and auditing systems are needed to comprehensively evaluate the impact of ASGs. Currently, quantification of AMU in small animals on an international scale, is based on tablet sales in individual nations [5,92] providing only an approximate indication of total usage in each species and no indication of the appropriateness of AMU or the degree of adherence to guidelines. The network on quantification of veterinary antimicrobial usage at herd level and analysis, communication and benchmarking to improve responsible usage (AACTING) has developed a centralized European resource that lists existing systems [93]. Monitoring systems have evolved in recent years and offer the potential to fill this information gap.

## 5. Practical Impact of Our Findings and Recommendations

This study provides an overview of the available AMU guidelines in small-animal veterinary practice and sheds light on the need for national guidance documents in multiple European countries. National bodies are encouraged to prioritize guideline development in small animals. This study provides a framework highlighting some of the fundamental stewardship principles that should be integral to future ASG and encourages guideline developers to adopt systematic and transparent methodologies. Academia, funding agencies and governmental institutions should prioritize future research to address deficiencies in the AMU evidence base. Guidelines must be sufficiently flexible and dynamic to be of value to the local population of users; engagement of relevant stakeholders can help address the specific needs of the target audience. Means to readily collate data relating to AMU in dogs and cats are required to enable auditing of the impact of ASGs. Demonstration of the long-term efficacy and cost-effectiveness of these interventions to combat AMR would validate the efforts of ASG developers and stimulate the further optimization of this field.

## Figures and Tables

**Figure 1 antibiotics-10-00409-f001:**
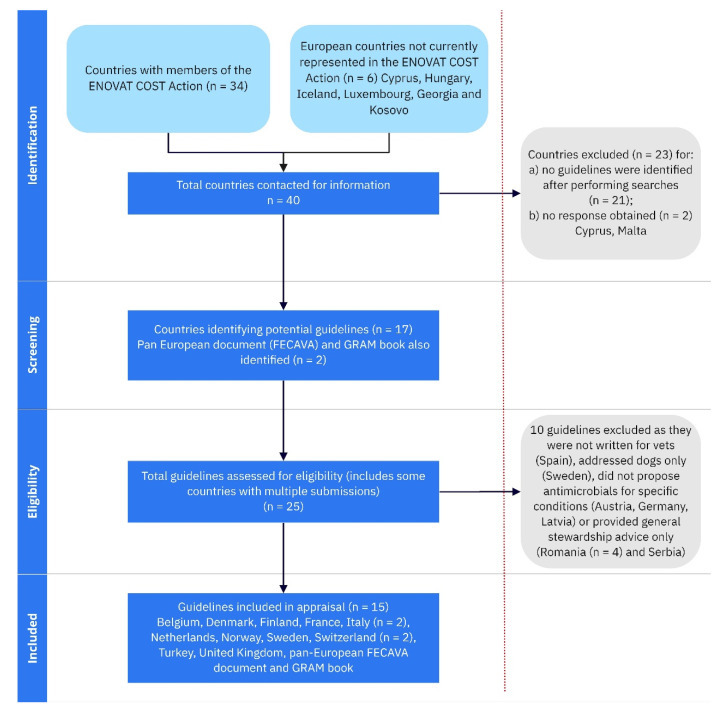
Flow diagram showing the identification, screening and selection of ASGs for detailed review. Excluded ASG are listed in Appendix A.

**Figure 2 antibiotics-10-00409-f002:**
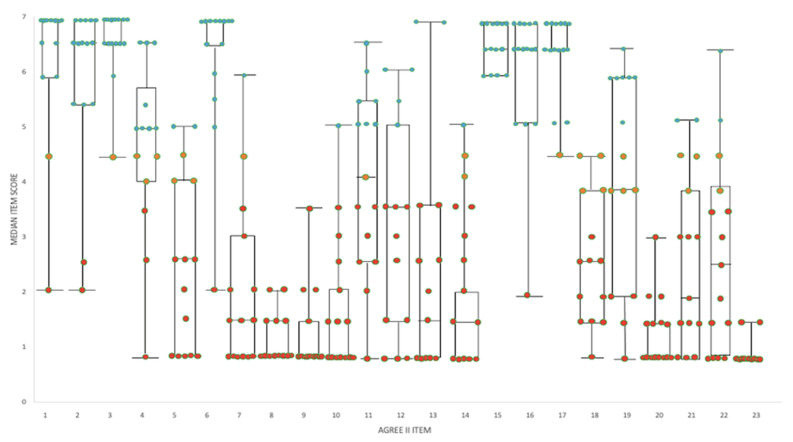
Box and whisker plot showing the median item score for all ASGs combined and range overlain with the median item scores for each individual ASG. Each ASG item score is represented by a color (green for median scores ≥ 5; orange for 4–5 and red for <4). See Table 3 for the description of each AGREE II item.

**Figure 3 antibiotics-10-00409-f003:**
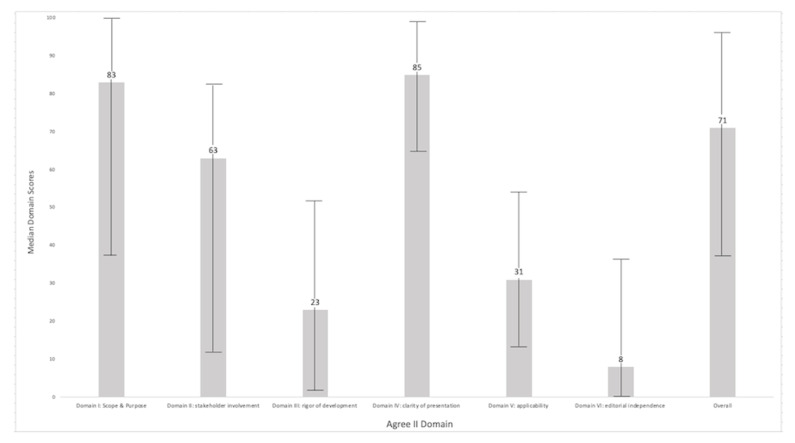
Median domain scores and range for all appraised ASGs.

**Table 1 antibiotics-10-00409-t001:** ASGs included in this study.

Country	Last Updated	Antimicrobial Stewardship Guidelines (ASGs)
Belgium	2020 [29]	Formularium Antimicrobial Consumption and Resistance in Animals (AMCRA)
Denmark	2018 [30]	Antibiotic Use Guidelines for Companion Animal Practice (2nd Edition) *
Finland	2018 [31]	Mikrobilääkkeiden käyttösuositukset eläinten tärkeimpiin tulehdus- ja tartuntatauteihin *
France	2017 [32]	Guide De Bonnes Pratiques Filière Animaux De Compagnie Fiches De Recommandations Pour Un Bon Usage Des Antibiotiques 2017
Italy	2017 [33]	Linee guida. Uso prudente dell’antibiotico negli animali da compagnia
	2017 [34]	Linee Guida sul corretto uso degli antibioticinella clinica del cane e del gatto
Netherlands	2017 [35]	Formularium gezelschapsdieren hond, kat en konijn
Norway	2014 [36]	Terapianbefaling: Bruk av antibakterielle midler til hund og katt
Sweden	2010 [37]	Guidelines for the clinical use of antibiotics in the treatment of dogs and cats *
Switzerland	2019 [38]	Therapieleitfaden für Tierärztinnen und Tierärzte—Hunde und Katzen
	2020 [39]	AntibioticScout.ch
Turkey	2017 [40]	Veteriner Hekimlikte Antibiyotikler (Pratik Bilgiler Rehberi) 2nd Edition
United Kingdom	2018 [41]	PROTECT ME poster *
FECAVA	2018 [42]	FECAVA Recommendations for Appropriate Antimicrobial Therapy *
GRAM book	2016 [43]	Guidance for the rational use of antimicrobials *

Those marked with an asterisk (*) are also available in English.

**Table 2 antibiotics-10-00409-t002:** Frequency of recommendations in ASGs.

Recommendation	Number of ASGs (n = 15)	Percentage of ASGs (%)
Antimicrobials are not indicated for management of:		
Acute diarrhea	15	100
Clean/elective surgical procedures	13	87
Feline lower urinary tract disease	11	73
subclinical bacteriuria	8	53
Non-antimicrobial therapeutic options described	14	93
Use topical medication instead of systemic medication where appropriate	15	100
Select narrow over broad-spectrum antimicrobials or encourage de-escalation to a narrower spectrum	13	87
Avoid certain antimicrobials reserved for human use only, e.g., vancomycin or carbapenems	12	80
Mention highest priority critically important antimicrobials (HPCIAs)	10	66
Tier antimicrobial suggestions (first line, second line)	13	87
Promote use of diagnostic techniques (cytology/culture) to identify putative bacteria	15	100
List common pathogens found in specific conditions	14	93
Monitor local antimicrobial resistance patterns	5	33
Audit/monitor individual/practice AMU	8	53

**Table 3 antibiotics-10-00409-t003:** Intraclass correlation coefficients, 95% confidence intervals and interpretation for each AGREE II item.

AGREE II Item		ICC	95% Confidence Interval		Correlation
1	The overall objective(s) of the guideline is (are) specifically described	0.64	0.21	0.86	Good
2	The clinical question(s) covered by the guideline is (are) specifically described	0.28	−0.57	0.73	Poor
3	The patients to whom the guideline is meant to apply are specifically described	0.52	−0.04	0.82	Fair
4	The guideline development group includes individuals from all the relevant professional groups	0.71	0.36	0.89	Good
5	The patients’ views and preferences have been sought	0.16	−0.82	0.68	Poor
6	The target users of the guideline are clearly defined	0.57	0.06	0.84	Fair
7	Systematic methods were used to search for evidence	0.42	0.26	0.78	Fair
8	The criteria for selecting the evidence are clearly described	−0.25	−1.9	0.54	Poor
9	The strengths and limitations of the body of evidence are clearly described	0.41	−0.28	0.78	Fair
10	The methods for formulating the recommendations are clearly described	0.56	0.03	0.83	Fair
11	The health benefits, side effects, and risks have been considered in formulating the recommendations	0.31	−0.49	0.74	Poor
12	There is an explicit link between the recommendations and the supporting evidence	0.58	0.08	0.84	Fair
13	The guideline has been externally reviewed by experts prior to its publication	0.70	0.34	0.89	Good
14	A procedure for updating the guideline is provided	0.42	−0.24	0.72	Fair
15	The recommendations are specific and unambiguous	0.52	−0.4	0.84	Fair
16	The different options for management of the condition are clearly presented	0.54	−0.01	0.83	Fair
17	Key recommendations are easily identifiable	0.15	−0.85	0.68	Poor
18	The guideline is supported with tools for application	0.39	−0.42	0.78	Poor
19	The potential organizational barriers in applying the recommendations have been discussed	0.42	−0.25	0.78	Fair
20	The potential cost implications of applying the recommendations have been considered	0.02	−1.14	0.63	Poor
21	The guideline presents key review criteria for monitoring and/or audit purposes	0.12	−0.91	0.67	Poor
22	The guideline is editorially independent from the funding body	0.42	−0.37	0.79	Fair
23	Conflicts of interest of guideline development members have been recorded	0.58	0.01	0.85	Fair

## Data Availability

The datasets used and/or analyzed during the current study are made available as Appendix A.

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
