# Peer review of "Overview and Evaluation of Existing Guidelines for Rational Antimicrobial Use in Small-Animal Veterinary Practice in Europe"

_antibiotics, 2021, doi:10.3390/antibiotics10040409_

Round 1
Reviewer 1 Report
The authors presented a research article aimed at analyzing the existing guidelines for the administration of antibiotics and other antimicrobials in small animals like cats and dogs. Overall, the authors have analyzed all the relevant literature on the topic, however, they take into account only the guidelines of European countries. The manuscript is well written, below are reported some comments that will improve the quality of the manuscript:
1) In the abstract section, please correct the following error: “...were asked to to identify...”;
2) The authors have taken into account all the relevant literature about ASG in cats and dogs. However, an important factor in the acquisition of MDR is the excessive use of antibiotics in livestock (cattle, pigs, sheep, etc.). I understand that these aspects are out of topic, however, the authors should briefly describe the impact of antibiotics used in livestock in the Introduction or Discussion section;
3) The discussion section should be significantly shortened;
4) In the following paragraph please add some information about the use of molecular tests: “All of the appraised guidelines encourage pathogen identification via cytology and/or bacterial culture to guide treatment decisions. Although bacterial culture with antimicrobial susceptibility testing (AST) is considered an essential stewardship tool [41, 42], studies suggest infrequent application in small animal practice in Europe often due to cost constraints [43-45].”;
5) Why do the authors not analyze the guidelines adopted in the USA, Canada and other extra-UE countries? Please clarify this critical aspect.
Reviewer 2 Report
This is a review describing the application and quality of Antibiotic stewardship guidelines (ASG) in the case of antibiotic use (AMU) in dogs and cats in European countries.
It is a very interesting and original manuscript. In my opinion, this topic has not yet been described in this range in the professional literature.
Due to the rise of bacterial resistance to antibiotics (AMR) and the possibility of losing the ability to treat bacterial infections, this is a very current topic. According to the One Health concept, the solution to the AMR issue must be comprehensive, including both the human and veterinary fields. I consider it appropriate to emphasize that One Health concept is an approach that recognizes that the health of people is closely connected to the health of animals and our shared environment. This isn't a new concept, but it has become more important in recent years.
I have the following comments on the article:
- I recommend replacing term Enterobacteriaceae with the term Enterobacterales and specifying that it means ESBL-positive Enterobacterales.
- The correct term is carbapenems, not carbopenems
- Although the manuscript is focused on the evaluation of existing ASG in Europe, I recommend supplementing the discussion with text describing the detection of multidrug-resistant bacteria in dogs and cats, especially MRSA, VRE (vancomycin-resistant enterococci), ESBL- and AmpC-positive Gram-negative bacteria.
- Are data available on antibiotic consumption in dogs and cats? If so, I recommend that you consider supplementing the discussion with this information.
Round 2
Reviewer 1 Report
In their point-by-point reply, the authors declared "
- The reviewer highlights a very important point. This study addresses the antimicrobial stewardship guidelines (ASG) available in countries represented in ENOVAT (and by extension Europe) only. This reflects the capacity to thoroughly seek guidelines in each country. The authors are aware of several additional documents that have been produced in the USA and Canada but note that no other ASG (for cats and dogs) have been included in the global repository of available guidelines for responsible use of antimicrobials in animal health that is available online: http://www.worldvet.org/news.php?item=417 (and included as reference 26). The authors were not able to thoroughly verify the availability of guidelines in these territories (absence of representatives) and therefore chose to focus our investigations on the ASG produced in Europe. It would be very interesting to extend this work more broadly in the future.".
Please add a brief statement in the conclusive remarks stating that guidelines or USA and Canada have not been considered in the manuscript because "the authors were not able to thoroughly verify the availability of guidelines in these territories (absence of representatives) and therefore chose to focus our investigations on the ASG produced in Europe."
Author Response
The authors thank the reviewer for their further review of this manuscript. The following statement has been added to the discussion section to reflect the point raised:
ASG from outside Europe have not been considered in this study; absent specific country representatives it was not possible to assuredly locate all ASG in these territories.